# Contribution of Epithelial Plasticity to Therapy Resistance

**DOI:** 10.3390/jcm8050676

**Published:** 2019-05-14

**Authors:** Patricia G. Santamaría, Gema Moreno-Bueno, Amparo Cano

**Affiliations:** 1Departamento de Bioquímica, Universidad Autónoma de Madrid (UAM), Instituto de Investigaciones Biomédicas ‘Alberto Sols’ (CSIC-UAM), IdiPAZ, c/ Arzobispo Morcillo 4, 28029 Madrid, Spain; gmoreno@iib.uam.es; 2Centro de Investigación Biomédica en Red de Cáncer (CIBERONC), c/ Monforte de Lemos 3-5, 28029 Madrid, Spain; 3Fundación MD Anderson Internacional, c/ Gómez Hemans 2, 28033 Madrid, Spain

**Keywords:** epithelial–mesenchymal transition, hybrid E/M states, plasticity, tumour heterogeneity, treatment resistance, immunotherapy scape

## Abstract

Therapy resistance is responsible for tumour recurrence and represents one of the major challenges in present oncology. Significant advances have been made in the understanding of the mechanisms underlying resistance to conventional and targeted therapies improving the clinical management of relapsed patients. Unfortunately, in too many cases, resistance reappears leading to a fatal outcome. The recent introduction of immunotherapy regimes has provided an unprecedented success in the treatment of specific cancer types; however, a good percentage of patients do not respond to immune-based treatments or ultimately become resistant. Cellular plasticity, cancer cell stemness and tumour heterogeneity have emerged as important determinants of treatment resistance. Epithelial-to-mesenchymal transition (EMT) is associated with resistance in many different cellular and preclinical models, although little evidence derives directly from clinical samples. The recognition of the presence in tumours of intermediate hybrid epithelial/mesenchymal states as the most likely manifestation of epithelial plasticity and their potential link to stemness and tumour heterogeneity, provide new clues to understanding resistance and could be exploited in the search for anti-resistance strategies. Here, recent evidence linking EMT/epithelial plasticity to resistance against conventional, targeted and immune therapy are summarized. In addition, future perspectives for related clinical approaches are also discussed.

## 1. Introduction

The emergence of therapy resistance is one of the main unsolved issues in present oncology and represents a major hurdle to defeating cancer.

Traditionally, two forms of tumour drug resistance, innate and acquired, have been considered responsible for tumour relapse either soon after initial treatment or even following several years of initial response and tumour shrinkage [1,2,3]. However, differences between both resistance definitions at mechanistic and molecular levels are somehow attenuated, especially after the accumulation of genomic and genetic data [2,4,5]. Accordingly, we will herein use the general term “treatment resistance” to refer to both types of resistance as well as to include resistance to diverse treatment regimens (radio, chemo or immune therapy, as well as targeted therapy).

A great effort has been made over the last two decades to unveil the molecular pathways responsible for therapy resistance. Such attempts have led to the identification of several molecular mechanisms involved in resistance to conventional (i.e., increased expression of anti-apoptotic or transporter proteins mediating multidrug resistance) and targeted therapies (like novel mutations bypassing specific inhibitors and/or activation of alternative signalling pathways) [6]. The introduction of next generation sequencing technology and the compilation of information regarding patients´ responses to different therapies for distinct tumour types is providing powerful information towards personalized treatments as well as facilitating the prediction of recurrences [2,5]. Notwithstanding, we still have an insufficient knowledge of the mechanisms driving tumour resistance.

Three concepts that emerged over the last years have provided novel insights into our present understanding of tumour resistance: cancer stem cells (CSCs), epithelial-to-mesenchymal transition (EMT) and intra-tumour heterogeneity [7], all of them briefly discussed hereafter. The discovery of CSCs soon led to the exposure of their increased resistance to chemo- and radiotherapy compared to non-CSCs in the same tumour (reviewed in Reference [8]). This point has been confirmed in different experimental situations in which conventional therapies were able to eliminate non-CSCs while slowly proliferating CSCs were unaffected [9,10,11,12]. The subsequent link between EMT and cancer stem-cell properties [13,14,15] set the grounds to associate EMT with therapy resistance [16]. In fact, original studies in tumour cell lines revealed that cells undergoing EMT achieve resistance to genotoxic stress mediated by conventional radio- and chemotherapy [17,18,19,20]. This was later confirmed using different therapeutic drugs reinforcing the link between CSCs, EMT and resistance [8,16,21,22,23]. While this hypothesis is highly attractive, we have partial knowledge on how these two (clinically relevant) programs, cancer cell stemness and EMT are interrelated. Clinical evidence linking them to resistance is limited, mostly due to the lack of appropriate in vivo models and scarce patient samples to perform comprehensive studies.

A major problem to regard EMT as a relevant process in cancer progression has been the difficulty to unambiguously detect its occurrence in most human tumours. Nonetheless, the recognition that intermediate EMT or hybrid epithelial/mesenchymal (E/M) states represent a likely situation in tumours [24,25,26,27], has brought novel insights on the relationship between epithelial plasticity and treatment resistance [28,29]. The well-established intra-tumour heterogeneity (referred to from now as tumour heterogeneity), considered as a common feature of most solid tumours, has also powered the notion that heterogeneity might be key for treatment resistance [30,31]. The relationship among heterogeneity, phenotypic plasticity and tumour resistance is thus emerging as a forefront of research in oncology [28].

Several recent outstanding articles have reviewed the association between EMT and cancer stemness in tumour progression and/or treatment resistance [8,23,24,25,28,29,32] and this will not be considered here in extent. In the present review, we will thus summarize recent evidence linking EMT/epithelial plasticity to therapy resistance as well as to immune scape. Future perspectives for refining predictive resistance biomarkers and novel clinical approaches will be also discussed.

## 2. Tumour Heterogeneity and Epithelial Plasticity: Traits Conferring Tumour Aggressiveness and Resistance

### 2.1. Tumour Heterogeneity Links Phenotypic Plasticity and Therapy Resistance

Nowadays, increased evidence supports that heterogeneous cancer cell populations with distinct phenotypic features constitute the majority of tumours [2,7]. The heterogeneity of tumours represents new challenges in oncology with particular relevance for precision medicine. It is presently considered that the resistance to diverse treatments in many cancer patients relies on tumour heterogeneity [8,30,31], highlighting the importance of understanding cancer heterogeneity for prognosis and therapy choices. Despite the partial knowledge of the driving mechanisms leading to tumour heterogeneity, several common features are starting to emerge, summarized as: (a) tumour heterogeneity can arise from both genetic and epigenetic mechanisms; (b) tumour cells are able to shift among several phenotypic states during tumour evolution; and (c) discrete populations of cancer stem cells within the tumour mass can give rise to hierarchically organized phenotypically distinct subpopulations [5,7,8,25,28,30,33]. An additional consequence of tumour heterogeneity is phenotypic plasticity, a determinant factor for cancer progression and therapy resistance [34]. As extensively reviewed elsewhere, phenotypic plasticity is being recognized in several tumour types, including breast and lung cancer, involving the acquisition of different histological traits and/or differentiation states that, at least in some cases, are associated with therapy resistance [25,28,34].

### 2.2. EMT and Epithelial Plasticity: A Short Story

The EMT program, classically defined as a coordinated cell and molecular process by which epithelial tumour cells progressively lose cell junctions and apical–basal polarity while acquiring mesenchymal capacities [24,35,36,37], is likely one of the major manifestations of phenotypic epithelial plasticity in tumours. However, it is worth recalling here that EMT is an essential biological program for early development as well as for establishing key embryonic structures such as, but not limited to, derivatives from the neural crest or the cardiac cushions [35]. Importantly, developmental EMT is a highly dynamic and transient process acting at discrete spatio-temporal contexts, thus requiring its quick reversal through a mesenchymal-to-epithelial transition (MET) process. Moreover, several rounds of EMT and MET occur during development and morphogenesis of several embryonic tissues [35] reinforcing the dynamic nature of epithelial plasticity in physiological contexts.

Epithelial-to-mesenchymal transition was originally described in vitro in epithelial renal cells cultured under different substrates and characterized by the loss of intercellular adhesion and acquisition of migratory and mesenchymal-like traits [38], and later demonstrated to occur in vivo during chicken-embryo gastrulation [39]. Since the beginning of the present century, a myriad of articles have described the acquisition of EMT by normal and malignant epithelial cells under different stimulus in culture, starting from the original identification of Snail transcription factor as an E-cadherin repressor and EMT-inducer [40,41] followed by the identification of additional EMT-transcription factors (EMT-TFs), such as Slug, zinc finger E-box binding homeobox 1 and 2 (ZEB1, ZEB2), Twist or the basic-loop-helix transcription factor 3 E47/TCF3 (reviewed in References [37,42,43]), presently considered as classical or core EMT-TFs (Table 1). A plethora of extrinsic signalling pathways regulating EMT in non-malignant and tumour epithelial cells have been deciphered, as well as different intrinsic regulatory mechanisms acting at post-transcriptional, post-translational and epigenetic levels [23,24,35,42,43,44] that will not be further discussed here. However, it is important to remark that in tumours, most of the present data support that EMT is essential for metastasis, in particular for initial invasion, as well as for intra- and extravasation [8,24,25,37,45,46], while the reverse MET seems to be required for colonisation and macro-metastasis generation at distant sites [47,48,49].

The relevance of the EMT program in cancer was strengthened by its connection to cancer cell stemness, initially described in normal and transformed human mammary cells in which EMT induction, by Snail or Twist expression, led to the acquisition of CSC markers, the ability to form mammospheres and tumour initiating capabilities [13,14]. Since then, the association of EMT and CSCs has been observed in several carcinoma types [8,13,15] supporting that the EMT program contributes to the self-renewing activity of CSCs in primary tumours and is potentially associated with therapeutic resistance [8,16,28]. Nevertheless, the association between EMT and CSCs during tumour progression and metastasis is not fully understood and, importantly, it might depend on particular tumour contexts [50], as exemplified by the non-classical EMT-TF paired related homeobox 1 (Prrx1) that represses CSC traits in triple-negative breast cancer (TNBC) cells while its silencing is required for metastatic colonization associated with the acquisition of stemness properties and an MET phenotype [47]. Further studies are undoubtedly needed to advance our knowledge on the EMT–CSCs link in connection to treatment resistance.

### 2.3. EMT In Vivo

In contrast to the overwhelming information on EMT in in vitro tumour cell models, the evidence for its occurrence in vivo is scarce. In fact, the relevance of EMT in human tumours has been widely questioned, in particular by pathologists, mainly because of the difficulty to detect full EMT inside tumours [51]. The generation of sophisticated genetically modified mouse models allowing EMT lineage tracing has provided convincing evidence for EMT occurrence in vivo (reviewed in References [26,27]). Some of these mouse cancer models combine the conditional deletion or activation of specific EMT-TFs and/or in vivo imaging. Information obtained from a genetic pancreatic mouse model showed that EMT cells (Zeb1^+^) appear in precursor pancreatic intraepithelial neoplasia (PanIN) lesions and are able to generate heterogeneous tumours containing E-cadherin^+^ and E-cadherin^−^ cells [52]. Early disseminated tumour cells with partial EMT and high metastatic potential were also detected in the MMTV-Her2 breast cancer model [53]. Another elegant system designed to track endogenous E-cadherin in MMTV-PyMT breast cancer mouse model combined with high-resolution intravital imaging allowed the identification of a subpopulation of cells undergoing EMT with invasive and metastatic properties, exposing as well high the intrinsic plasticity of EMT cells at metastatic sites [54], similar to results obtained using a different breast cancer model [55]. Lineage tracing in a Notch-p53-based colorectal cancer (CRC) mouse model also provided evidence for invasive cells exhibiting a gradient of epithelial and mesenchymal phenotypes [56]. In addition, several mouse models based in the genetic manipulation of EMT-TFs have reported the implication of Snail or Twist1 in EMT induction in PyMT-breast cancer or skin squamous cell carcinomas (SCC), respectively [48,57]. Interestingly, these two studies demonstrated that both Snail and Twist1 are transiently expressed and needed for initial EMT-mediated invasion and dissemination, but their corresponding expression should be shut down for metastatic outgrowth in each tumour context.

Nonetheless, two recent studies using EMT lineage tracing and conditional deletion of specific EMT-TFs (Snail or Twist) in breast and pancreatic cancer mouse models, respectively, concluded that EMT is not required for metastasis but indeed contributes to chemoresistance [58,59]. As previously discussed by others [23,60,61], these unexpected and contradictory results can be explained either by the use of non-specific Cre-driver lines and/or by the redundant actions of specific EMT-TFs. In support of the later hypothesis, further analyses in the same pancreatic mouse cancer model by Zheng et al. [59] showed that the deletion of other EMT-TF such as Zeb1 strongly decreased lung metastasis, demonstrating that EMT is indeed required for distant metastasis [62]. Importantly, this particular study also contributed to establish that the role played by EMT-TFs can be redundant in a context-dependent fashion and that distinct EMT-TFs, or specific combinations of them also defined as “EMT-TF code”, can be required to drive distant metastasis in different tumour settings [62,63,64].

### 2.4. Intermediate EMT States in Tumours: Novel Insights on Epithelial Plasticity

Present data suggest that EMT in vivo can be considered not as a binary process (epithelial versus mesenchymal states) but rather as a combination of several cellular states in which a spectrum of epithelial and mesenchymal traits can coexist [23,28,29] (Table 1). Hybrid E/M states can, theoretically, be associated with plasticity programs endowing tumour cells with a metastable state with the ability to rapidly respond to microenvironmental signals in either direction (i.e., towards an epithelial or a mesenchymal state) [24,25,26]. Accumulated data from cell lines and preclinical models are starting to provide evidence about the existence intermediate or hybrid E/M states in different tumour settings (see Reference [29] for a recent review). Computational and mathematical modelling analyses have been recently used to design hypothetical models of hybrid E/M states that are being tested to prove for tumour cell metastable or stable phenotypes as well as for their link to therapy resistance [65,66,67,68].

Initial insights for intermediate EMT states came from different studies showing the coexistence of epithelial and mesenchymal markers in tumour cell lines under various experimental conditions as well as in several tumour series (reviewed in Reference [27]). Of particular interest are the immunohistochemical analyses where the co-expression of epithelial (E-cadherin or keratins) and mesenchymal (N-cadherin or vimentin) markers was initially characterized in different breast tumour series [69,70]. Following studies characterized basal-like breast tumours as co-expressing vimentin and cytokeratins [71], and further identified an EMT immunohistochemical signature as specifically associated with basal-like breast tumours as well as the coexistence of epithelial and mesenchymal markers in the epithelial component of breast carcinosarcomas [72]. This later study represents one of the first hints pointing to the existence of a partial EMT in human tumours. However, until very recently, direct evidence for different intermediate E/M state occurrence in tumours was lacking. A recent study in a genetic mouse model of skin SCC has allowed to identify and isolate six intrinsic cell subpopulations with different combinations of epithelial and mesenchymal markers that define distinct EMT transition states in vivo [73]. This study shows that intermediate E/M populations are more metastatic than full M populations; in addition, intermediate E/M subpopulations exhibit a high degree of cell plasticity being able to switch into one another in secondary tumours [73]. Remarkably, and in agreement with those observations, depletion of the EMT-TF Zeb1 in pancreatic mouse tumours halts tumour cells in a stable epithelial state, losing their stemness and metastatic properties together with the ability to induce EMT upon transforming growth factor beta (TGFβ) signalling [62]. Overall, these recent studies support that epithelial plasticity conferred by intermediate E/M states is highly relevant for metastatic dissemination.

The influence of hybrid E/M states to treatment resistance is poorly understood, although several theoretical and experimental studies are starting to provide information of this relevant aspect, and they will be discussed below.

## 3. Evidence Linking EMT to Treatment Resistance

Work in cancer cellular models and patient samples, mostly analysing gene expression profiles that can be associated with therapy response, have allowed to establish a link between the gene expression associated with an EMT program and the development of therapeutic resistance [8]. The underlying mechanisms have been extensively reviewed [8,23,28] being essentially related to increased ability of EMT cells to avoid apoptosis induced by most standard anti-cancer drugs, implementation of mechanisms mediating drug detoxification and expression of immunosuppressive and immunoevasive molecules to avoid attack by the immune system. While the impact of hybrid E/M states to tumour aggressiveness is starting to be elucidated [29], several signalling pathways and molecular mechanisms are emerging as potential common traits of EMT and treatment resistance. For instance, a cellular signalling network consistently linked to EMT-mediated drug resistance across different cancer types is conveyed by the AXL tyrosine kinase receptor (known as AXL) that alters mitogen-activated protein kinase (MAPK)/ERK and phosphoinositide 3 kinase (PI3K)/Akt signalling pathways favouring proliferation, survival, migration and invasion [74]. Briefly, the AXL relationship to EMT (either as effector or inducer) has been explored in different tumour types, being associated with metastasis and drug resistance and, thus worse prognosis in patients (recently reviewed in Reference [74]). Additionally, AXL has been implicated in engaging other receptor tyrosine kinases (RTKs) and their downstream signalling in ovarian cancer [75] or epidermal growth factor receptor (EGFR) signalling in TNBC [76], which seem to be relevant in EMT cancer cells and associated resistance to RTK-targeted therapies.

In addition, a cellular adaptive mechanism known as the unfolded protein response (UPR) is activated to cope with the endoplasmic reticulum stress resulting from tumour progression (reviewed in References [77,78,79]). In cancer cells, UPR activation reduces the pro-apoptotic effects of several chemotherapeutic drugs and favours drug detoxifying mechanisms. Unfolded protein response activation has been suggested to uphold EMT, becoming both programs’ allies in cancer initiation and progression (reviewed in another chapter in this special issue [80]) and contributing to cellular adaptive mechanisms responsible for chemotherapy resistance.

Tumour microenvironments can also have an important role in EMT and treatment resistance. Numerous studies have demonstrated that EMT programs in cancer cells are elicited by an array of signals originating from the different components of the tumour stroma [24,42,44]. Among them, cancer associated fibroblasts (CAFs), tumour associated macrophages (TAMs), infiltrating T-lymphocytes and myeloid-derived suppressor cells (MDSCs) can play prominent roles in the paracrine regulation of EMT induction, mainly mediated by TGFβ, tumor necrosis factor alpha (TNFα) or interleukin 6 (IL-6) secretion, among other cytokines and growth factors (reviewed in References [8,23]). The cytokine TGFβ is perhaps the most potent EMT-inducing signal in many different tumour contexts and indeed its secretion by CAFs and/or TAMs leads to EMT induction in breast cancer and hepatocarcinoma, among other tumour cells [81,82,83,84], while IL-6 secretion by CAFs has been associated to EMT-mediated resistance in non-small cell lung cancer (NSCLC) [85]. Interestingly, TGFβ has been proposed as a determinant of metastatic dissemination in CRC models [86] and poor prognosis CRC subtypes share a gene program driven by stromal TGFβ that seems to be associated with treatment resistance [87]. Recently, stromal TGFβ has been linked to immune evasion in CRC and urothelial tumours [88,89], although the potential connection to EMT induction has not been directly addressed. At present, the influence of the microenvironment in regulating intermediate E/M states and their association with therapeutic resistance is basically unknown, but it can be speculated that paracrine signals from CAFs, TAMs and other stromal components are relevant players in the dynamic regulation of epithelial plasticity in cancer progression.

### 3.1. Studies on Tumour Cell Lines

There are many examples in different cancer settings in which the expression of one or several core EMT-TFs is linked to increased resistance to radio- and chemotherapy as well as to targeted therapy [19,20,90,91,92,93,94,95]. Moreover, resistant tumour cells in culture frequently exhibit a mesenchymal phenotype [8] supporting the EMT involvement in therapy resistance. We will next discuss recent literature exemplifying such a link in several cellular cancer models.

#### 3.1.1. Lung Cancer

In 2013, an EMT gene signature comprising 76 genes allowed to classify NSCLC cell lines as epithelial or mesenchymal, the latter expressing higher levels of *ZEB1*, *vimentin* or *AXL* [96]. This EMT signature also predicted the sensitivity of patient-derived NSCLC cell lines to different drugs, being the mesenchymal ones more resistant to the EGFR inhibitors (EGFRi) (erlotinib, gefitinib), as well as to PI3K inhibitors and common cytotoxic chemotherapies such as docetaxel or paclitaxel. However, the classification of a cell line as mesenchymal was not linked to widespread drug resistance since they were more sensitive to cisplatin or gemcitabine than the epithelial ones [96]. These studies and others have led to regard EMT as crucial for the generation of NSCLC resistance to EGFRi and the molecular mechanisms involved have been recently reviewed elsewhere [97]. In brief, cell stemness traits, anti-apoptotic signalling and chromatin remodelling imposed by EMT-TFs would cooperate to promote therapy resistance in NSCLC. As an example, the repression of the pro-apoptotic protein Bcl-2-like protein 11 (BIM) by ZEB1 seen in mesenchymal NSCLC cells is accountable for the increased resistance to EGFRi treatments [98].

Also, recent results link the presence of cells with hybrid E/M features in NSCLC cell lines to EGFRi resistance, increased sphere-forming ability and ZEB1 expression [99]. Other studies supporting the association of intermediate E/M phenotypes and resistance have described the expression of integrin beta4 (ITGB4), a proposed marker of the E/M state [100], in CSCs of NSCLC [101], although additional studies are required to sustain this connection.

In small cell lung cancer (SCLC) cells, the activation of the Met receptor with hepatocyte growth factor (HGF) induces a mesenchymal phenotype involving enhanced expression of EMT-TFs such as Snail together with increased invasion, tumorigenesis and chemoresistance to etoposide in xenograft assays [102]. Chemosensitivity could be restored in the presence of a Met inhibitor [102] further supporting the link between EMT and resistance in lung cancer.

#### 3.1.2. Breast Cancer

Normal and transformed human mammary epithelial cells induced through an EMT by inhibition of E-cadherin expression or Twist overexpression are resistant to paclitaxel and doxorubicin, common chemotherapeutic drugs, whereas breast cancer cells with EMT traits show an increased sensitivity to paclitaxel [21,94]. A recent report has found links between EMT and endocrine therapy resistance in luminal breast cancer. Estrogen receptor alpha gene (*ESR1*) fusion proteins found in luminal tumours are responsible for endocrine therapy refractory disease [103]. In fact, when expressed in breast cancer cell lines, these functional fusion proteins (to the C-terminal sequence of the Hippo pathway coactivator yes-associated protein 1 (YAP1) or protocadherin 11X PCDH11X) promote an estrogen-independent activation of an EMT gene signature, Snail upregulation and E-cadherin downregulation resulting in increased migration in vitro and lung metastasis in xenograft models [103].

In basal-like breast cancer MDA-MB-231 cells, an early study showed that Snail confers resistance to the standard chemotherapeutic agents docetaxel and gemcitabine [20]. More recently, the ubiquitin editing enzyme A20 has been shown to ubiquitinate and stabilize Snail in basal-like breast cancer cells, favouring TGFβ-induced EMT and lung colonization of orthotopic tumours [104]. In vitro, A20 expression is associated with enhanced breast cancer cell viability upon doxorubicin and docetaxel treatment [104]. Another recent report has described that the deubiquitinase (DUB), USP27X, regulates Snail stability in MDA-MB-231 cells [105]. In the absence of USP27X, Snail is degraded, and the growth of xenograft tumours is delayed as well as Snail-mediated metastasis and resistance to cisplatin. The authors found a positive correlation of Snail and USP27X expression in TNBC patient-derived xenografts (PDX), although their status in relapsed patients requires further investigation. Besides USP27X, DUB3 has also been shown to stabilize Snail favouring EMT-related invasion, migration and metastasis of xenograft tumours whereas DUB3 levels in breast cancer patients are associated with metastatic progression and quicker relapse [106]. In addition, in basal-like breast cancer cells, DUB3 stabilizes Slug and Twist1 [107], suggesting that EMT-TF stabilization by preventing proteasome degradation is an important contributor to EMT and associated roles in tumour progression.

Recent studies also imply that hybrid E/M sates can more efficiently favour metastasis and therapy resistance than a complete mesenchymal state in breast cancer cells (reviewed in Reference [29]). The mechanistic pathways underlying chemotherapy resistance associated with E/M states in basal-like TNBC is being deciphered, as exemplified by ITGB4 expression regulated by ZEB1 and its downstream target Tap63a [100], but further studies are required.

#### 3.1.3. Ovarian Cancer

In ovarian cancer, upregulation of Snail and Slug has been detected in cisplatin resistant cell lines [92]. Moreover, both EMT-TFs are associated with radio and chemoresistance by p53-dependent pro-survival signalling and regulation of stemness in this tumour context [108]. In response to cisplatin, doxorubicin or paclitaxel, ovarian cancer cells with CSCs traits are selected in vitro, characterized by a mesenchymal phenotype with downregulated transcript levels of *E-cadherin* and *occludin*, and higher transcript levels of *fibronectin*, *Snail*, *N-cadherin*, *TWIST*, *ZEB1* and *ZEB2* [109]. These cells, displaying chemokine receptor type 4 (CXCR4) surface markers, show enhanced migration, invasion and tumour-forming ability, and higher expression of ATP binding cassette subfamily B member 1 (ABCB1), a protein involved in acquiring multidrug resistance [109]. Besides, high-grade serous ovarian patients display higher levels of CXCR4 expression in their circulating tumour cells (CTCs), while targeting CXCR4 in preclinical models has been shown to decrease peritoneal dissemination in part by blocking EMT [110].

#### 3.1.4. Prostate Cancer

Epithelial plasticity, noticed in some prostate cancer cell lines, was previously linked to cell stemness, tumour aggressiveness and metastatic potential [49,111,112], but few studies have so far analysed the relationship of EMT and hybrid E/M states to resistance in prostate cancer cells [29]. Recently, EMT has been involved in resistance to radiotherapy of prostate cancer cells. Lysyl oxidase-like 2 (LOXL2), a protein promoting EMT [113], is upregulated both in prostate cancer cell lines and radioresistant patient samples and seems to be responsible for radiotherapy resistance in prostate cancer cells and derived xenografts by implementing EMT [114]. Another study showed that chemoresistance to the taxane cabazitaxel was relieved by antiandrogen-mediated reversion of EMT towards MET in preclinical models such as PDX and genetic mouse models of advanced prostate cancer [115].

#### 3.1.5. CRC

In CRC, a recent report depicts a novel mechanism involving EMT in progression and drug resistance [116]. While looking for substrates of the F-Box E3-ubiquitin ligase FBXW7 in intestinal stem cells, the authors find FBXW7 binds and ubiquitinates ZEB2 upon glycogen synthase kinase 3 beta (GSK3β) phosphorylation. In fact, in mouse and human CRC cell lines, ZEB2 induces EMT and is responsible for increased metastasis upon tail vein or orthotopic cell injection in nude mice. Also, ZEB2 is linked to the expression of stemness markers in colonospheres and organoids as well as increased drug resistance in CRC cell lines [116].

#### 3.1.6. Melanoma

Malignant melanoma, an invasive tumour characterized by high genetic and phenotypic heterogeneity, commonly presents drug resistance. In melanoma, cell plasticity has been associated with resistance by means of ZEB1 reprogramming. Interestingly, low levels of *MITF*, a key melanocyte lineage TF, and high *ZEB1* levels are associated with B-Raf protein kinase inhibitor (BRAFi) resistance both in vitro and in tumour samples [117]. Using melanoma cellular models, Richard and co-authors [117] demonstrated that ZEB1 expression promotes the upregulation of stemness markers, increased tumorigenic potential and resistance to BRAFi. These observations are in agreement with the previous description that in vitro resistance to BRAFi is accompanied by loss of melanocyte inducing transcription factor (MITF), E-cadherin and *ZEB2* expression and upregulation of *ZEB1* and *TWIST*, linked as well to enhanced invasion [118]. Altogether, these data support the role of a mesenchymal phenotypic switching in melanoma related to dedifferentiation, invasiveness and drug resistance.

A recent report has linked microenvironmental cues such as nutrient starvation to translational reprogramming and therapeutic resistance in melanoma [119]. Upon glutamine starvation, melanoma cells downregulate MITF through UPR activation (eIF2α-ATF4), resulting in increased invasiveness linked to ZEB1, N-cadherin and fibronectin upregulation and Slug downregulation. Since ZEB1 and Slug have been previously involved in the phenotypic switch responsible for malignant melanoma [120], UPR activation is thus linked to the melanoma plasticity required for invasiveness. Indeed, ATF4 also correlates with higher AXL expression, a mediator of BRAF and MEK inhibitor (MAPKi) resistance in melanoma [121] also associated with anti-PD-1 therapy resistance [122]. These recent studies also indicate that MAPKi resistance in melanoma cells involves a mesenchymal signature, with decreased MITF and increased ZEB2 and Slug expression [121], suggestive of epigenetic regulatory mechanisms [122].

#### 3.1.7. Glioblastoma

The EMT-TF ZEB1 has been involved in glioblastoma formation, invasion and chemoresistance in cellular models [123]. The proposed molecular mechanism involved EMT-induced cancer cell stemness and ZEB1-miR200 dependent upregulation of *O*-6-Methylguanine DNA methyltransferase (MGMT), responsible for resistance to the standard of care drug temozolomide (TMZ) [123]. Further studies to characterize ZEB1 involvement in glioblastoma by expression profiling and chromatin binding site analysis in CSCs have revealed that ZEB1 activates and represses distinct sets of target genes implementing a complex genetic program similar to EMT [124].

### 3.2. Computational Modelling Analyses on Epithelial Plasticity and Tumour Resistance

The presence of heterogeneous phenotypes within tumours before treatment and their plausible plastic transition across different E/M states might be useful to inform the outcome and, thus select therapies targeting particular phenotypes. To gain insight into phenotypic plasticity involvement in tumour heterogeneity and drug resistance, Gupta and co-workers [125] developed a method to mark the few cells within a breast cancer cell line that contain subpopulations of both epithelial and mesenchymal phenotypes. Expansion of this cell line originated clones composed of epithelial or mesenchymal cells or a mixture of both. In addition to concluding that phenotypic plasticity is inherited, these authors used these in vitro data gathered from these marked clones to perform computational simulations of the outcome of tumours containing mixed clones with different E/M phenotypes upon drug treatment. They modelled the effect of different chemotherapeutic drugs, selectively targeting epithelial or mesenchymal phenotypes, in different therapy regimens and concluded that the most efficient treatment is the combination therapy, repeated alternation of drugs targeting epithelial or mesenchymal phenotypes compared to monotherapy or sequential therapy [125].

Other computational modelling has provided an EMT metric to predict the extent of the EMT status, either epithelial, mesenchymal or hybrid E/M, of a given transcriptomic sample, aiming at being clinically informative [67]. Based on data from gene expression profiles, this EMT score has been validated in cancer cell lines with known EMT phenotypes and it provides valuable information regarding EMT score and survival as well as relapse upon treatment.

### 3.3. Studies on Mouse Models

At present, few reports have analysed EMT and treatment resistance in cancer mouse models, aside from analyses of xenografted tumour cell lines in immune-deficient mice.

A role for EMT in in vivo chemoresistance has been backed recently by two studies in mouse cancer models based on EMT downregulation [58,59]. Pancreatic mouse tumours generated after knocking down Snail or Twist were more sensitive to gemcitabine, the standard of care drug [59]. Additionally, in an EMT lineage tracing mouse model of breast cancer, EMT inhibition by miR-200 overexpression, which directly targeted *Zeb1* and *Zeb2*, the tumour resistance to the chemotherapy drug cyclophosphamide was abrogated [58]. Although the limitations of both studies regarding EMT contribution to metastasis have since been argued as already mentioned [46,60,61], they provided significant in vivo preclinical evidence of EMT involvement in chemoresistance.

## 4. Insights on EMT and Treatment Resistance in the Clinical Setting

The association of EMT and epithelial plasticity with resistance in the clinical context is not completely understood. As discussed above, EMT can provide tumour cells with the abilities to escape apoptosis, anoikis and senescence, among other traits and, thus confer treatment resistance in several preclinical models [8,28,29,126], but direct proof for this mechanistic link in tumours is still missing.

Nevertheless, different evidence supports the impact of EMT on treatment resistance in human tumours, such as the studies in which gene expression profiles from tumour samples are correlated to the clinical behaviour of treated patients. Some of these studies have resulted in the identification of several EMT-related gene expression signatures strongly associated with conventional and targeted therapy resistance, particularly in breast cancer and NSCLC [96,127,128].

### 4.1. EMT and Resistance to Conventional and Targeted Therapy

Epithelial plasticity and CSCs connection to treatment resistance is being recognised in particular tumour types [8,28,129,130] from where some common hints are emerging. In fact, it has been reported that in several tumour types, and particularly in pancreatic cancer, a minimal tumour fraction of resistant undifferentiated CSCs exhibit a spindle-shaped appearance typical of an EMT phenotype [8,29,126]. In this sense, both the expression of specific EMT-TFs and the acquisition of a mesenchymal or undifferentiated phenotype within tumours have been related to an adverse therapeutic outcome [28]. Nonetheless, few studies have so far focused on deciphering the clinical correlation between EMT and resistance (see Table 2 for examples) beyond lung cancer (see Reference [97] for a recent review).

Regarding conventional chemotherapy, one example is seen in prostate cancer, in which resistance to docetaxel has been associated with EMT; presenting resistant tumours lower E-cadherin expression and decreased miR-200 levels [133]. In ovarian cancer patients, expression profiling analyses identified a molecular signature differentially expressed in chemoresistant and chemosensitive patients [132]. Importantly, *TWIST1* expression was significantly higher in ovarian cancer patients with poor therapeutic response upon platinum regimens (Table 2), and the authors proposed that chemoresistance was due to Twist1-mediated inhibition of apoptosis [132]. Moreover, biopsies from relapsed squamous cell lung carcinoma (SCLC) patients treated with platinum and etoposide showed enhanced levels of EMT-related mesenchymal proteins such as Snail, vimentin and the extracellular matrix protein SPARC and decreased expression of E-cadherin [102] (Table 2). Nonetheless, and in contrast to in vitro approaches, the association between EMT and resistance is not easy to characterize in the clinical context, probably because cancer patients normally receive complex therapeutic regimens, which would mask the relevance of cell plasticity to resistance against specific chemotherapeutic agents [137]. Despite this fact, the link between hybrid E/M states and chemoresistance has been established in a relevant study in human breast cancer, where the presence of CTCs, showing hybrid E/M traits, is associated with patients that exhibit increased resistance to combined chemotherapeutic and targeted agents [138].

On the other hand, the role of EMT in radioresistance has been studied in some tumour subtypes such as prostate cancer [134] (Table 2). Among different molecular mechanisms, including resistance to anoikis or PI3K/Akt pathway signalling activation, EMT induction has been observed in relapsed prostate cancer patients after radiotherapy. In this tumour context, radiation decreases E-cadherin expression by a mechanism dependent on Snail expression concomitant with N-cadherin and vimentin upregulation [139]. In addition, EMT-induced radioresistance is associated with a dramatic increase of the DNA repair gene poly (ADP-ribose) polymerase 1 (*PARP-1*) supporting that, in these radioresistant patients, treatment with PARP inhibitors might represent a new therapeutic approach worth being evaluated [139,140,141].

As already mentioned, the depiction of EMT in clinical samples has been linked to the lack of response to some particular targeted therapies, mostly by the association of specific EMT-TFs expression with resistance [28]. As examples, ZEB2 and Slug are overexpressed in MAPKi-resistant melanomas [121], while ZEB1 expression is linked to BRAFi resistance in melanoma patients [117], reduced response to the EGFRi erlotinib in NSCLC [96] and to TMZ in glioblastomas [123] (Table 2). Also, a correlation between Snail expression and resistance to erlotinib in head and neck squamous cell carcinoma (HNSCC) was described, at least in preclinical models [142]. Overall, these data gathered from clinical samples suggest that a dedifferentiation state (Table 2), modulated by the activation of an EMT program, appears as a key determinant for therapeutic resistance [28,29], although a thorough understanding of the mechanisms operating in cancer patients has not been achieved yet.

Despite numerous efforts to develop therapeutic strategies that directly or indirectly interfere with the EMT program (i.e., by blocking the secretion of EMT inducers, inhibiting EMT-TFs or targeting specific EMT-induced intracellular pathways [28]), none of them have so far benefitted patients in terms of resistance reversion. In fact, it was observed that these potential EMT-targeted therapies could trigger the activation of alternative pathways that behave as compensatory resistance mechanisms [143]. Likely, this could be related to the hybrid E/M phenotypes present in tumours as discussed above, even though EMT blockage remains nowadays a challenge in oncology and there are currently more than 30 clinical trials being conducted and focused on EMT reversion in many cancers, not only using chemo and/or targeted therapies (i.e., NCT01990196, NCT00769483 and NCT03509779) but also radiotherapy (i.e., NCT03660319 and NCT02913859) [144].

### 4.2. EMT and Immunotherapy: A Further Link to Immune Evasion

Immunotherapy approaches, including monoclonal antibodies, checkpoint inhibitors, therapeutic vaccines and adoptive cell transfer, have emerged as a promising therapeutic strategy for cancer treatment, particularly in melanoma, lung, bladder, NSCLC and HNSCC tumours. Immunotherapy, currently focused on immuno-inhibitors targeting the interaction of programmed cell death 1 (PD-1) with PD ligand 1 (PD-L1) or cytotoxic T-lymphocyte antigen 4 (CTLA-4) (reviewed in References [145,146]) provides clear benefit for some cancer patients. However, it is yet a challenge due to the high number of patients not profiting from immune-based treatments, some of them actually resistant to immunotherapy [147].

Tumour cells undergoing EMT have been shown to circumvent immune surveillance and become refractory to immune-based therapies [23,29,148]. Although not completely understood, recent evidence is providing some clues into the molecular bases of the link between EMT/epithelial plasticity and immune evasion [148]. The detected CTCs positive for PD-L1 show spindle-like morphology resembling intermediate EMT phenotypes [149]. Moreover, in NSCLC recurrent patients under treatment with the PD-L1 inhibitor nivulumab, the co-expression of some EMT markers (N-Cadherin and vimentin) and PD-L1 was detected in their CTCs [136] (Table 2) suggesting that the EMT phenotype of PD-L1 CTCs might identify those NSCLC patients not responding to immune therapy. Furthermore, immunotherapy response deficiency, also known as tumour immune escape, has been partially associated with the upregulation of some EMT-TFs [148]. It has been recently noted that the suppression of anti-tumour immunity, affecting CD8+ T cells in the tumour microenvironment, could be due to the miR-200/ZEB1 axis, which directly regulates the expression of PD-L1 in lung cancer [150]. The association of EMT with an immunosuppressive phenotype has also been observed in melanoma, pancreatic and breast cancer patients [148]. Epithelial breast carcinoma cells expressing high levels of Snail showed significant susceptibility to CTL-mediated lysis reduction [151] and this has been related to the activation of pro-survival autophagy [152]. In colon cancer biopsies, PD-L1 expression has been detected in tumour buds, located at the invasive fronts of tumours and thought to be formed by cancer cells undergoing EMT [153]. In fact, PD-L1 expression in tumour buds positively correlates with ZEB1 and ZEB2 expression, suggesting that an EMT program might be linked to immune evasion by upregulating PD-L1 in CRC patients [153].

Besides, exome and transcriptome sequencing data obtained from melanoma samples, suggest that tumours unresponsive to anti-PD-L1 therapy display a gene signature related to mesenchymal phenotypes, also induced upon treatment with MAPKi and present in residual tumours treated with MAPKi, suggesting that common mesenchymal features are associated with resistance to targeted and immune therapies [122].

Furthermore, an EMT transcriptional score was measured in many tumour subtypes [128] and the immunotherapy response was better in those patients presenting tumours with luminal (epithelial) phenotype than in those patients with basal (undifferentiated or mesenchymal like) phenotype [154]. Furthermore, a high EMT score has been related to immune marker expression (i.e., PD-L1, PD-L2, and CTLA-4, among others) [155], and PD-L1 association with EMT was confirmed in lung adenocarcinomas [156] and HNSCC [157]. In NSCLC, the mesenchymal tumours showed an increase in tumour infiltrating lymphocytes (TILs) and regulatory T (Treg) cells [156]. A high EMT score in NSCLC tumours was also correlated with expression of the immune modulator CD276, regarded as a new prognostic marker for overall survival. Additionally, mesenchymal NSCLC tumour subclones, but not those with epithelial phenotype, presented increased ability to resist lysis-induced by natural killer (NK) cells [158]. Some studies also suggest that tumour cells undergoing EMT show a significant reduction in the MHC class I receptor [148], which participates in the activation of additional lytic cell death mediated by NK cells [148,159].

Further studies are required to precisely decipher the immune scape mechanisms occurring in cancer. Nevertheless, some hints point to an important contribution of EMT/epithelial plasticity to immune escape as well as a to the potential utility of EMT assessment in patients as a predictive biomarker for immune therapy selection.

## 5. Novel Perspectives for Targeting EMT-Mediated Resistance

Hitherto, the development of new therapeutic approaches to minimise EMT-induced treatment resistance in cancer is essential. The strategies for targeting EMT and resistance are particularly directed at the reversion of EMT and/or dedifferentiation programs. Various recent approaches will be briefly described in this section. With the objective to selectively kill cancer stem cells, several attempts have been implemented to find molecules targeting cells undergoing EMT [21,94,160]. In this sense, standard chemotherapy has been demonstrated to be unable of killing cells undergoing EMT in several carcinoma cellular models. Gupta and co-authors [21] exploited this fact in experimentally induced EMT in untransformed and transformed human mammary epithelial cells by downregulating E-cadherin and observed an increase in their resistance to several established chemotherapeutic drugs. In particular, treatment with paclitaxel selected resistant mesenchymal and migratory cells displaying markers associated with human mammary CSCs as well as showing increased tumour seeding and metastasis in xenografts [21]. This resistance model was used in a high-throughput screen of chemical compounds leading to the identification of salinomycin, a potassium ionophore, with cytotoxic activity on EMT cells [21]. Other high-throughput screening identified PKCα inhibitors able to eliminate human mammary cells that underwent EMT [161], further supporting that EMT can confer vulnerabilities in order to tackle tumour resistance.

Dedifferentiation compelled by EMT activation has also been linked to multidrug resistance in breast cancer cell lines [162]. In vitro, the overexpression of Twist increases resistance to chemotherapeutic drugs through enhanced ability to cope with oxidative stress by the activation of the UPR. Indeed, the inhibition of UPR prior to treatment with chemotherapeutic drugs leads to a delayed growth of basal breast xenografted tumours [162].

On the other hand, EMT-endowed plasticity can be exploited to favour drug-induced MET or transdifferentiation. With the aim of identify compounds inducing MET in mesenchymal breast cancer cells, a screening for drugs able to induce E-cadherin transcription uncovered two classical activators of PKA (cholera toxin and forskolin) as inducers of the epithelial state [163]. The epithelial-derived breast cancer cells upon treatment with PKA activators lose their stemness properties and develop increased sensitivity to conventional chemotherapy. Mechanistically, the phenotypic reversion depends on the PKA substrate H3K9 histone demethylase PHF2 that derepresses epithelial gene expression by epigenetic mechanisms [163]. This finding is also in agreement with the previous discovery that the HDAC inhibitor mocetinostat reverts ZEB1-associated resistance of cancer cells [164] lending further support to the potential use of epigenetic modulators to revert EMT-associated resistance [26].

In fact, transdifferentiation has been a long-standing goal in anti-cancer treatment. However, in the context of EMT/epithelial plasticity, reversion to a MET state can represent a double-edged strategy because of the association of MET with metastasis colonization at distant sites [47,48]. Therefore, complete eradication of EMT tumour cells, and particularly of those acquiring intermediate E/M states, can be envisioned as a steadier strategy against tumour metastasis and resistance. A recent elegant study has exploited the plasticity of intermediate E/M cancer cells to force their transdifferentiation into post-replicative adipocytes [165]. Murine breast cancer cells forced to undergo EMT were induced to irreversibly become adipocytes in vitro with a cocktail containing rosiglitazone, a peroxisome proliferator activated receptor gamma (PPARγ) agonist, and bone morphogenetic protein 2 (BMP2), being the cells in hybrid E/M states the most sensitive ones. During the cancer cell transdifferentiation, the expression of EMT-related and proliferation genes was reduced to become cell cycle arrested adipocytes. Indeed, Snail downregulation and ZEB1 upregulation were necessary, whereas MEK/ERK signalling had a negative effect towards adipogenesis of EMT cells. In vivo, the treatment with a MEKi (trametinib) and rosiglitazone, so-called adipogenesis therapy, reduced the growth and metastatic colonization of breast orthotopic tumours formed by injecting MDA-MD-231 LM basal-like breast cancer cells, similar to the results obtained in a TNBC-derived PDXs [165]. Interestingly, the analysis of the tumours upon adipogenesis therapy showed that the cancer-derived adipocytes were located at the tumour rims, where the expected EMT cells responsible for invasiveness and metastasis are located [166]. Furthermore, the tumours presented a rather differentiated phenotype with upregulation of E-cadherin expression. Since EMT cells were able to transdifferentiate into other mesenchymal cell types such as osteoblasts and chondrocytes depending on the differentiation protocol used [165], the therapeutic exploitation of the plasticity that EMT grants should be further explored in other solid tumours (Figure 1).

In melanoma, studies using single-cell sequencing showed that drug resistance is achieved through epigenetic reprogramming [167]. The authors found that rare cells within the bulk population expressed high levels of resistance markers (such as platelet-derived growth factor receptor (PDGFR) or AXL) in pre-treated cultures, giving then rise to a resistant population upon BRAFi treatment. They concluded that a transient pre-drug pre-resistant state allows tumour cells to readily acquire stable resistance when exposed to a drug and they suggest that this reprogramming is accompanied by phenotypic changes [167], which are reminiscent of EMT even if this point was not directly analysed. Interestingly as well, a recent study shows that minimal residual disease in melanoma is associated with cell and spatial heterogeneity and identifies transcriptional programs associated with neural crest cell stemness as key drivers of resistance to established targeted inhibitors [168]. Since a phenotypic switch associated with the change in expression of specific EMT-TF in melanoma progression has been previously identified [120], it is tempting to speculate that such phenotypic plasticity can also be exploited for epigenetic reprogramming associated with drug resistance in melanoma.

## 6. Conclusions

The EMT program and epithelial plasticity have been associated with resistance to chemo, radio and targeted therapies, as well as to novel immune-based treatments in different tumour contexts. Mechanistic insights behind such relationships are starting to emerge from experimental cellular and preclinical tumour models. Although evidence in clinical settings is still scarce, the recent appreciation that intermediate or hybrid E/M states might represent a more likely situation in tumours, together with their potential involvement in tumour heterogeneity and stemness, are providing new opportunities to expand our understanding of the contribution of epithelial plasticity to treatment resistance. As depicted in Figure 1, the attained knowledge will provide additional means to design therapeutic strategies aimed at reverting resistance by targeting epithelial plasticity and eliminating E/M cells similar to the induction of irreversible differentiated cell states.

## Figures and Tables

**Figure 1 jcm-08-00676-f001:**
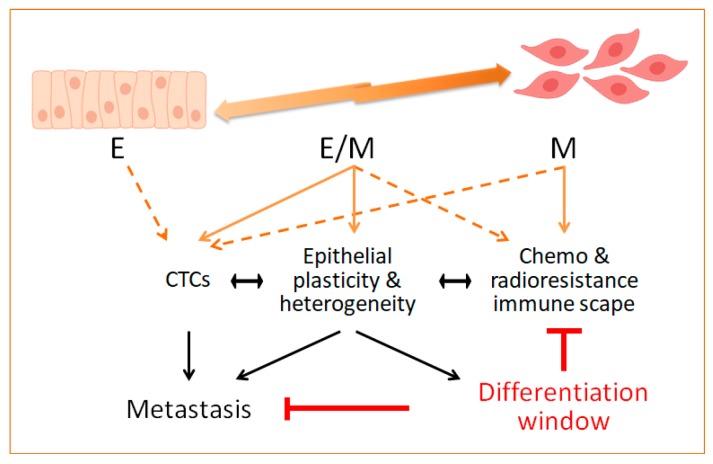
Tumours are formed by heterogeneous and phenotypically diverse cancer cell populations. During tumour progression, epithelial cells lose their apical–basal polarity and acquire mesenchymal traits through the Epithelial-to-mesenchymal transition (EMT) program. In vivo, EMT generates a wide spectrum of cellular phenotypes from epithelial (E) to mesenchymal (M) phenotypes, accompanied by gain of migratory and invasive abilities. Cells in hybrid E/M states give raise to heterogeneous populations, some endowed with stem cell-like features. These metastable E/M cells, able to rapidly adapt to changes in the tumour microenvironment, could ultimately be responsible for tumour resistance to anti-cancer therapy and immune scape. Indeed, hybrid E/M cells are associated with tumour progression, metastatic dissemination and tumour recurrence since they thrive in hostile situations due to their inherent plasticity. Circulating tumour cells (CTCs) isolated from patients display E/M traits and, in some tumour types, are considered crucial for metastatic colonization. Since treatment resistance and metastasis are the main consequences of cancer progression, drugs aimed at exploiting epithelial plasticity by promoting a cell irreversible differentiation state might constitute a successful anti-cancer strategy.

**Table 1 jcm-08-00676-t001:** EMT-TFs and main characteristics associated with the EMT program.

EMT State	Epithelial (E)	Hybrid E/M	Mesenchymal (M)
Morphology	Apical–basal polarity, cells attached to each other and to extracellular matrix (ECM)	Partial loss of cell–cell and cell–ECM attachment, epithelioid shape	Front–rear polarity, elongated shape, detached cells
Markers	E-cadherin, claudins, occludins, cytokeratins *	Co-expression of E and M markers: E-cadherin, cytokeratins *, vimentin	N-cadherin, vimentin, fibronectin, matrix metalloproteinases (MMPs), fibrillar collagens
Associated functional traits	Restrained motility, regulated proliferation	Motility, invasion, stemness, dissemination, metastasis, immune evasion, therapy resistance	
Core EMT-TFs:	Snail & Slug, ZEB1 & ZEB2, Twist, E47/TCF3		

* Cytokeratins (Krts) such as Krt8/18 are commonly detected in E states whereas Krt5/14 in E/M states. EMT: epithelial-to-mesenchymal transition; EMT-TFs: EMT-transcription factors.

**Table 2 jcm-08-00676-t002:** Examples of the contribution of epithelial plasticity to treatment resistance in cancer patients.

Specific Therapy	Tumour Subtype	Status *	Phenotype #	Reference(s)
***Chemotherapy***				
Platinum/etoposide	SCLC ^1^	Clinical	Undifferentiated	[102]
Taxanes	NSCLC ^1^	Clinical	Differentiated	[131]
Cisplatin	Ovarian	Clinical	Undifferentiated	[132]
Docetaxel/Cabazitaxel	Prostate	Clinical/preclinical	Differentiated	[115,133]
***Radiotherapy***				
Radiotherapy	Prostate	Clinical	Differentiated	[114,134]
***Targeted therapy***				
Temazolomide ^2^	Glioblastoma ^MGMT-met^	Clinical	Undifferentiated	[123]
Erlotinib ^3^/temsirolimus ^4^	HNSCC ^1^	Clinical	Differentiated	[135]
Cobimetinib ^3^	Melanoma	Preclinical	Undifferentiated	[117]
Vemurafenib ^3^	Melanoma	Clinical/preclinical	Undifferentiated	[117]
Erlotinib/gefitinib ^3^	NSCLC ^EGFR-mut^	Preclinical	Undifferentiated	[96]
***Immunotherapy***				
Nivulumab ^5^	NSCLC (CTCs) ^1^	Clinical	Undifferentiated	[136]

* Clinical: study on patient tumour samples; Preclinical: study using preclinical models (like patient derived xenografts (PDXs)). # Estimated according to the tumour cell morphology. ^1^ SCLC: squamous cell lung cancer; HNSCC: head and neck squamous cell carcinoma; NSCLC: non-small cell lung cancer; CTCs: circulating tumour cells. ^2^ Alkylating cytotoxic prodrug. ^3^ Epidermal growth factor receptor/mitogen activated protein kinase/B-Raf protein kinase (EGFR/MAPK/BRAF) inhibitors. ^4^ mammalian target of rapamycin (mTOR) inhibitor. ^5^ monoclonal antibody against PD ligand 1 (mAb αPD-L1).

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
