# Peer review of "Contribution of Epithelial Plasticity to Therapy Resistance"

_jcm, 2019, doi:10.3390/jcm8050676_

Reviewer 1 Report

Epithelial plasticity contribution to therapy resistance

The authors Santamaría et. al. have reviewed the literature on the overall contribution of EMT and hybrid E/M states in tumor progression and therapy resistance. They discussed these effects based on in-vitro studies and patient clinical data with respect to therapy resistance in solid tumors. They furthered the topic by also including the information on computational analyses. Lastly, they also discussed the relevance of EMT in context of immune therapy resistance and conclude by suggesting a therapy regimen that targets the EMT and E/M states in addition to the conventional therapy regimens.

Overall, the review is written very precisely and is very well structured in including most up-to date information on the concept with respect to different solid cancers. Including few more changes can further add to the quality of this review.

1)      Minor English editing.

2)      A summary on role of microenvironmental influence on EMT and E/M states in regulating therapy resistance can be included to add to the relevant information provided in this review.

3)      Please include a reference for line 93-94.  Page 2.

4)      Section 3.1.2 Line 228. Please specify normal and transformed mammary epithelial cells.

5)      Including a summary table for section 3, EMT and E/M states in therapy resistance in all different cancer would be suitable.

6)      The title can be modified to “Contribution of epithelial plasticity to therapy resistance” would be better.

Author Response

1.     English has been reviewed and edited.

2.     A summary of the microenvironmental influence on EMT and E/M states has been included at the end of the introduction to section 3 (see page 6, lines 235-253). A comment on the scarce information regarding the influence of the microenvironmental cues in therapy resistance has been also included.

3.     The requested reference for line 93-94, page 2 (present line 95, page 3) has been included

4.     “Normal and transformed mammary epithelial cells” has been specified in section 3.1.2 (present line 285, page 7).

5.     Table 2 (original Table 1) has been now modified (see page 10) to include the tumour cell morphology (differentiated or undifferentiated) as described in the cited papers regarding their studies. This is because EMT or E/M status in relation to therapy is not specified in these particular studies.

6.     Title has been modified as suggested “Contribution of epithelial plasticity to therapy resistance”.

Reviewer 2 Report

Resistance to treatment is indeed a major hurdle for the treatment of cancers. This review provided relevant references, in vitro, in vivo and clinically, supporting the role of EMT in treatment resistance, not only to conventional anti-cancer drugs but also to targeted treatment and immunotherapy.

For better clarity for readers not in the field, the authors could provide a table listing markers and TFs related  to EMT (E, hybrid, and M) which were mentioned several times in the manuscript  

·         Section 2.3 EMT in vivo model, there is only a short statement referring to the convincing in vivo model for EMT  with a citation to 2 reviews. The authors could expand more in detail regarding how the evidence was convincing.

·         The author could provide a short paragraph about stem cell and epithelial plasticity. Because this is mentioned  in the figure 1, think they can expand a  bit

Author Response

-        A new table (Table 1) displaying the main characteristics of E, E/M and M states including morphology, markers, associated traits and core EMT-TFs has been incorporated (see page 4). We will like to remark that a global association of specific EMT-TFs to either E/M or full M state is presently unknown and the examples cited in the literature relate to cellular and/or mouse models of particular tumour types. Thus, a generalization of this specific aspect cannot yet be made. For the same reason, we only include very general markers for E and E/M states as specifics will depend on the epithelial and carcinoma context. This is particular relevant in the case of cytokeratins.

-        Section 2.3. “EMT in vivo” has been extended to describe some of the most relevant genetic models validating EMT in vivo (page 4-5, lines 145-174). This is not an exhaustive description, as the available models have been recently reviewed (see refs. included in this new paragraph), but we think it can provide basic information to non-specialist readers.

-        A short paragraph succinctly describing the connection between stem cells and EMT has been included at the end of section 2.2. “EMT and epithelial plasticity: a short story” (page 4, lines 128-140).